# DEEP SINGLE IMAGE MANIPULATION

## ABSTRACT

Image manipulation has attracted much research over the years due to the popularity and commercial importance of the task. In recent years, deep neural network methods have been proposed for many image manipulation tasks. A major issue with deep methods is the need to train on large amounts of data from the same distribution as the target image, whereas collecting datasets encompassing the entire distribution of images is impossible. In this paper, we demonstrate that simply training a conditional adversarial generator on the single target image is sufficient for performing complex image manipulations. We find that the key for enabling single image training is extensive augmentation of the input image and provide a novel augmentation method. Our network learns to map between a primitive representation of the image (e.g. edges and segmentation) to the image itself. At manipulation time, our generator allows for making general image changes by modifying the primitive input representation and mapping it through the network. We extensively evaluate our method and find that it provides remarkable performance.

## 1 INTRODUCTION

Images capture a scene at a specific point in time. Viewers often wish the scene had been different e.g. that objects were arranged differently. Due to the popularity of this task, it has been the focus of much research and also of many companies and products e.g. Instagram and Photoshop. Deep learning methods have significantly boosted performance of image manipulation methods for which large training datasets can be obtained e.g. super-resolution or facial inpainting. User-captured photographs follow a long tailed distribution. Some classes of photographs are very common e.g. faces or cars. On the other hand a large proportion of photographs capture a rare object class or configuration. Training deep learning methods that capture the entire distribution images can be very hard, particularly for generative models that are slow and tricky to train. Training models on just the target image is emerging as an alternative to training deep models on large image datasets. Although this is counter-intuitive as deep learning methods typically require many training samples, single-image methods have recently demonstrated some promising results.

In this paper, we introduce a novel method for training deep conditional generative models from a single image. The objective differs from popular single-image methods e.g. Deep Image Prior and SinGAN that focus on unconditional image manipulation. The training image is first represented with a primitive representation, which can be unsupervised (an edge map, unsupervised segmentation), supervised (segmentation map, landmarks) or a combination of both. We use a standard adversarial conditional image mapping network to learn to map between the primitive representation and the image. In order to extend the training set (which simply consists of a single image), we perform extensive augmentations. The choice of augmentation method makes a significant difference to the method's performance. We find the crop-and-flip augmentations typically used in conditional image generation are insufficient for providing a sufficiently rich training distribution. We propose to use a thin-plate-spline (TPS) augmentation method and show that it is key to the success of our method. After training, we are able to perform challenging image manipulation tasks by modifying the primitive representation. Our method is evaluated extensively and displays remarkable results.

Our contributions in this paper:

1. A general purpose approach for training conditional generators from a single image.

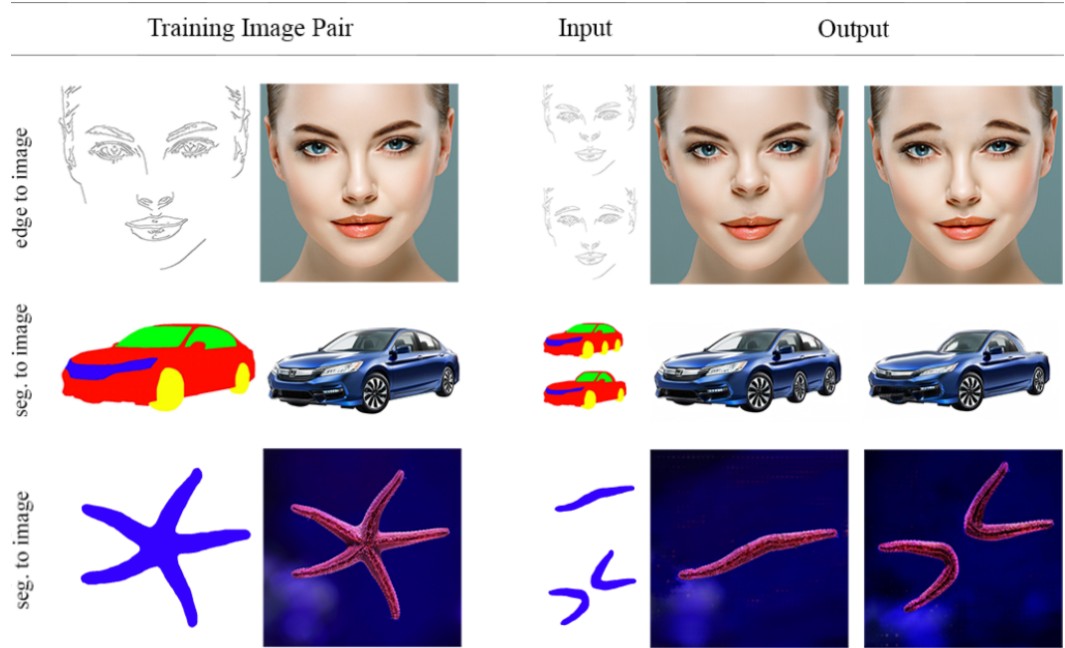

Figure 1: Results produced by our model. The model was trained on a single training pair (first and the second columns). The third column shows the inputs to the trained model at inference time. First row- (left) lifting the nose, (right) flipping the eyebrows. Second row- (left) adding a wheel, (right) conversion to a sports car. Third row - modifying the shape of the starfish.

2. Proposing a TPS-based augmentation for conditional image generation, and demonstrating its importance for single image training.

3. A primitive representation allowing concurrent low and high-level image editing.

4. Extensive evaluations showing remarkable visual performance, and the introduction of a new benchmark enabling quantitative evaluation of single image manipulation.

## 2   RELATED WORK

*Classical image manipulation methods:* Image manipulation has attracted research for decades from the image processing, computational photography and graphics communities. It would not be possible to survey the scope of this corpus of work in this paper. We refer the reader to the book by Szeliski (2010) for an extensive survey, and to the Photoshop software for a practical collection of image processing methods. A few notable image manipulation techniques include: Poisson Image Editing (Pérez et al., 2003), Seam Carving (Avidan & Shamir, 2007), PatchMatch (Barnes et al.) and ShiftMap (Pritch et al., 2007). Learning a high-resolution parametric function between a primitive image representation and a photo-realistic image was very challenging for pre-deep learning methods.

*Deep conditional generative models:* Image-to-image translation maps images from a source domain to a target domain, while preserving the semantic and geometric content of the input images. Most image-to-image translation methods use Generative Adversarial Networks (GANs) (Goodfellow et al., 2014) that are used in two main scenarios: i) unsupervised image translation between domains (Zhu et al., 2017a; Kim et al., 2017; Liu et al., 2017; Choi et al., 2018) ii) serving as a perceptual image loss function (Isola et al., 2017; Wang et al., 2017; Zhu et al., 2017b). Existing methods for image-to-image translation require many labeled image pairs. Several methods e.g Dekel et al. (2017), are carefully designed for image manipulation, however they require large datasets which are mainly available for faces or interiors and cannot be applied to the long-tail of images.

*Non-standard augmentations:* Conditional generation models typically use crop and flip augmentations. Classification models also use chromatic and noise augmentation. Recently, methods have

been devised for learning augmentation for classification tasks e.g. AutoAugment (Cubuk et al., 2018). Mounsaveng et al. (2019) learned warping fields for augmenting classification networks. Thin-plate-spline transformation have been used in the medical domain e.g. Tang et al. (2019), but they are used for training on large datasets rather than a single sample. Zhao et al. (2019) learned augmentations for training segmentation networks from a single annotated 3D medical scan (using a technique similar to Kanazawa et al. (2016)) however they require a large unlabeled dataset of similar scans which is not available in our setting. TPS has also been used as a way of parametrizing warps for learning dense correspondences between images e.g. Han et al. (2018) and Lee et al. (2020).

*Learning from a single image:* Although most deep learning works use large datasets, seminal works showed that single image training is effective in some settings. Asano et al. (2019) showed that a single image can be used to learn deep features. Limited work has been done on training image generators from a single-image. Deep Image Prior (Ulyanov et al., 2018), retargeting (Shocher et al., 2018a) and super-resolution (Shocher et al., 2018b) from a single image. Recently, the seminal work of Shaham et al. (2019), presented a general approach for single image generative model training. However these methods are mainly focused on unconditional image manipulations (or very limited conditional manipulation). Our method is focused on conditional manipulations and is able to affect significantly more elaborate changes to images including to large objects in the scene.

## 3 DeepSIM: Learning Conditional Generators from a Single Image

We propose DeepSIM, a conditional generative adversarial network (cGAN) for learning to map from a primitive representation (e.g. edges, segmentation) to the image. The approach has several objectives: i) single image training ii) fidelity - the output should reflect the primitive representation iii) appearance - the output image should appear to come from the same distribution as the training image. We present a novel augmentation method that allows standard cGAN architectures to achieve these objectives. The particular type of augmentations used is critical as they provide a prior over the data. Having a good prior is important as the training set consists of merely a single image.

### 3.1 Model:

Our model design follows standard practice for cGAN models (particularly Pix2PixHD (Wang et al., 2017)). Let us denote our training image pair $(x, y)$ where $y \in \mathbb{R}^{d_x \times d_y \times 3}$ is the input image ($d_x$ and $d_y$ are the number of rows and columns) and $x \in \mathbb{R}^{d_x \times d_y \times d_p}$ is the corresponding image primitive ($d_p$ is the number of channels in the image primitive). We learn a generator network $G : \mathbb{R}^{d_x \times d_y \times d_p} \to \mathbb{R}^{d_x \times d_y \times 3}$, which learns to map input image primitive $x$ to the generated image $G(x)$. The fidelity of the result is measured using the VGG perceptual loss $\ell_{perc} : (\mathbb{R}^{d_x \times d_y \times 3}, \mathbb{R}^{d_x \times d_y \times 3}) \to \mathbb{R}$ (Johnson et al., 2016) , which compares the differences between two images using a set of activations extracted from each image using a VGG network pre-trained on the ImageNet dataset (we follow the implementation in Wang et al. (2017)). We therefore write the reconstruction loss $\ell_{rec}$:

$$\ell_{rec}(x, y; G) = \ell_{perc}(G(x), y) \tag{1}$$

*Conditional GAN loss:* Following standard practice, we add an adversarial loss which measures the ability of a discriminator to differentiate between the (primitive, generated image) pair $(x, G(x))$ and the (primitive, true image) pair $(x, y)$. The conditional discriminator $D : (\mathbb{R}^{d_x \times d_y \times d_p}, \mathbb{R}^{d_x \times d_y \times 3}) \to [0, 1]$ is implemented using a deep classifier which maps a pair of primitive and corresponding image into the probability of the two being a groundtruth primitive-image pair. $D$ is trained adversarially against the generator network $G$. The loss of the discriminator ($\ell_{adv}$) is:

$$\ell_{adv}(x, y; D, G) = \log(D(x, y)) + \log(1 - D(x, G(x))) \tag{2}$$

The combined loss $\ell_{total}$ is the sum of the reconstruction and adversarial losses, weighted by a constant $\alpha$:

$$\ell_{total}(x, y; D, G) = \ell_{rec}(x, y; D) + \alpha \cdot \ell_{adv}(x, y; D, G) \tag{3}$$

| Primitive | Image | Primitive after TPS | Image after TPS |
|---|---|---|---|

Table 1: A source image and a primitive representation under a random TPS warp. *Also see App. H.*

## 3.2 AUGMENTATIONS:

When large datasets exist, finding the generator $G$ and conditional discriminator $D$ that optimize $\ell_{total}$ under the empirical data distribution can result in a strong generator $G$. However, as we only have a single image pair $(x, y)$, this formulation severely overfits. This has the negative consequence of $G$ not being able to generalize to new primitive inputs. In order to generalize to new primitive images, the size of the training dataset needs to be artificially increased so as to cover the range of expected primitives. Conditional generative models typically use simple crop and shift augmentations. We will later show (Sec. 4) that this simple augmentation strategy however will not generalize to primitive images with non-trivial changes. We first construct a non-smooth warp $t$ by selecting a set of pixel locations $(i, j)$ and perturbing them by a uniformly distributed shifts $(d_i, d_j) \sim (U(-\frac{d_x}{L}, \frac{d_x}{L}), U(-\frac{d_y}{L}, \frac{d_y}{L}))$ s.t. $t(i, j) = (i + d_i, j + d_j)$ (we use $L = 10$). In order to have a simple transformation, we do not do this for every pixel but only for a set of 4 equispaced pixels along each axis of the image. We learn a warp $f : (\mathbb{R}^{d_x}, \mathbb{R}^{d_y}) \to (\mathbb{R}^{d_x}, \mathbb{R}^{d_y})$ using a thin-plate-spline (TPS) to smooth the random transformation $t$ into a more realistic warp $(i', j') = f(i, j)$. $f$ is parameterized using a thin-plate spline (see Duchon (1977) for more details). The TPS loss $L_{TPS}$ consists of two terms, a loss that encourages $f$ to be similar to $t$ while the other encourages smoothness:

$$\min_{\omega} L_{TPS} = \sum_{i,j} \|t(i, j) - f(i, j)\|^2 + \lambda \int \int f_{xx} + f_{yy} + 2f_{xy} dx dy \tag{4}$$

Where $f_{xx}, f_{xy}, f_{yy}$ denote the second order partial derivatives of $f$. $\lambda$ is a regularization constant determining the smoothness of the warp. The optimization over the warp $f$ can be performed very efficiently e.g. Donato & Belongie (2002). We denote the distribution of random TPS that can be generated using the above procedure as $\Omega$.

## 3.3 OPTIMIZATION:

At training time, we sample random warps from the distribution $\Omega$. We use each random warp $f \sim \Omega$ to transform both the input primitive $x$ and image $y$ to create a new training pair $(f(x), f(y))$ (where we denote $f(x)(i, j) = x(i', j')$ where $(i', j') = f(i, j)$). We optimize the generator and discriminator adversarially to minimize the expectation of the loss $\ell_{total}$ under the empirical distribution of random TPS warps:

$$D', G' = \min_G \max_D \mathbb{E}_{f \sim \Omega} \ell_{total}(f(x), f(y); D, G) \tag{5}$$

We implemented the conditional GAN using the Pix2PixHD architecture. We kept the same hyperparameters as in the official repository except changing the number of iterations to 16000.

## 3.4 PRIMITIVE IMAGES:

In order to be able to edit the image, we condition our generator on a representation of the image that we call the image primitive. The required properties of the image primitives are being able to precisely specify the required output image and the ease of manipulation by image editor. These two objectives are in conflict, although the most precise representation of the edited image is the edited image itself, this level of manipulation is impossible to achieve by a human editor, in fact simplifying this representation is the very motivation for this work. Two standard image primitives used by previous conditional generators are the edge representation of the image and the semantic

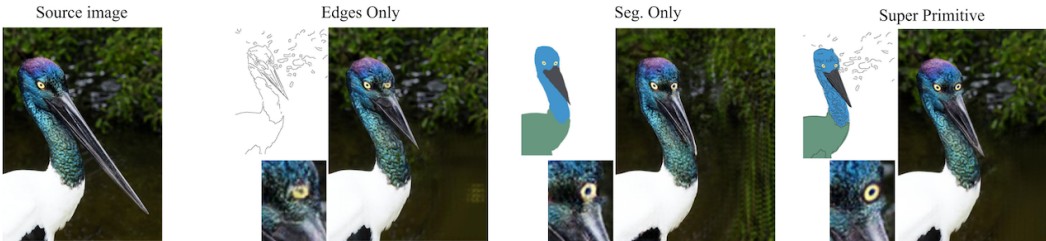

Figure 2: Results on three different image primitives (the leftmost column shows the source image, then each column demonstrate the result of our model when trained on the specified primitive). We manipulated the image primitives, adding a right eye, changing the point of view and shortening the beak. Our results are presented next to each manipulated primitive. Our SP performed best on high-level changes (e.g. the eye), and low-level changes (e.g. the background).

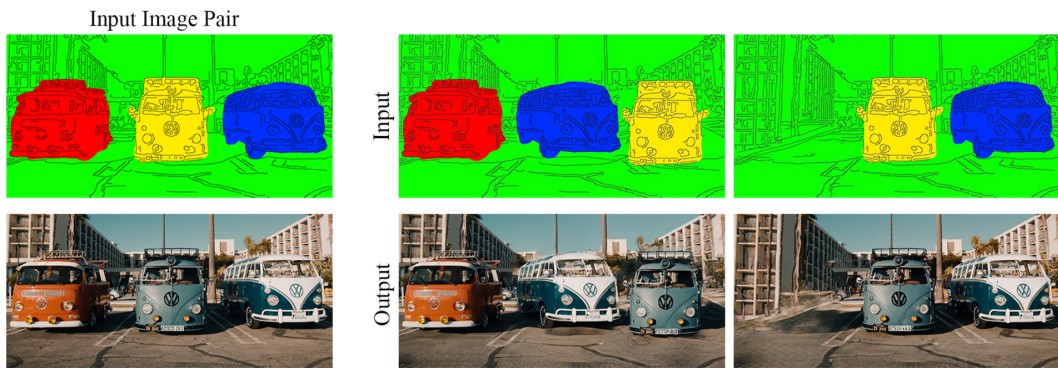

Figure 3: Results of our SP on challenging image manipulation tasks. left) the primitive-image pair used to train our method. center) switching the positions between the two rightmost cars. right) removing the leftmost car and inpainting the background. In both cases our method was able to synthesize very attractive output images. *See App. A for many more results.*

instance/segmentation map of the image. Segmentation maps provide information on the high-level properties of the image, but give less guidance on the fine-details. Edge maps provide the opposite trade-off. To achieve the best of both worlds, we propose a novel primitive, which includes both the edge and segmentation maps combined together (we dub this representation - "super primitive"). The advantages of this primitive representation will be shown in Sec. 5. Note that although Pix2Pix-HD proposes using a combination of segmentation maps and the edges of the instance maps, their representation is different from ours. The instance segmentation edges merely provide a method for overcoming the indexing issue present in instance maps, but do not provide the high-resolution of fine details in the edge maps as in our proposed approach. Our editing procedure is illustrated in App. B.

## 4 EXPERIMENTS

### 4.1 QUALITATIVE EVALUATION

We present many results of our method in the main paper and the appendix. In Fig. 1, our method is able to generate very high resolution images from single image training. In the top row we are able to perform fine changes to the facial images from edge primitives e.g. raising the nose and flipping the eyebrows. In the second row, we show complex shape transformations by using segmentation primitives. Our method was able to add a third wheel to the car and convert its shape into a sports car. This shows the power of the segmentation primitive, enabling major changes to the shape using simple operations. On the bottom row, we show that our method can perform free-form changes, completely changing the shape of an image while preserving fine texture.

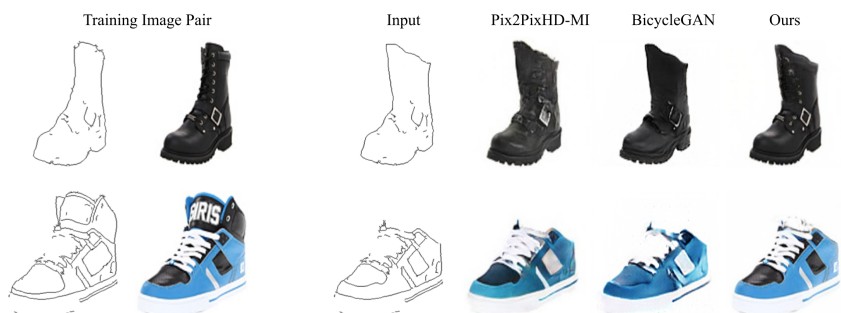

Figure 4: Edges-to-image results. columns 1, 2 show the edges and images used for training. Column 3 shows the edges used as input at inference time. We can see that Pix2PixHD-MI cannot generate the correct shoe as there is not enough guidance. BicycleGAN has sufficient guidance but cannot reproduce the correct details. Our approach generates images of high quality and fidelity.

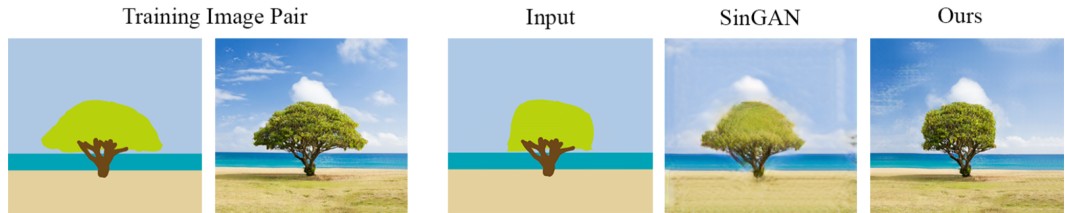

Figure 5: Paint-to-image results. Two leftmost columns show the input paint that was created manually and the input image. The third column shows the modified paint image used as input to the trained models, the result by SinGAN was generated using the authors' best practice. Our method generates a novel tree corresponding to the segmentation map with high fidelity. Note that SinGAN is not designed for conditional generation, hence the weak results.

In Fig. 4, we demonstrate the advantages of our single image model by comparing to two models that were trained on a large dataset. We present edges-to-image translation results on two different shoes. We can see that Pix2PixHD-MI (Pix2PixHD that was trained on the entire edge2shoes dataset, where "MI" is an acronym for "Multi Image") is unable to capture the correct identity of the shoes as there are multiple possibilities for the appearance of the shoe given the edge image. BicycleGAN is able to take as input both the edge map and guidance for the appearance (style) of the required shoe. Although it is able to capture the general colors of the required shoe, it is unable to capture the fine details of the shoes (e.g. shoe laces and buckles). We believe that the loss of information is a general disadvantage of training on large datasets, a general mapping function becomes less specialized and therefore less accurate on individual images.

In Fig. 5, we present results on a paint-to-image task. Our method was trained to map from a rough paint image to an image of a tree, while SinGAN was trained using the authors' best practice. We can see that SinGAN outputs an image which is more similar to the paint than a photorealistic image. Our method is able to change the shape of the tree to correspond to the paint while keeping the appearance of the tree and background as in the training image.

## 4.2 QUANTITATIVE EVALUATION

As previous single image generators have mostly operated on unconditional generation, there are no established datasets and metrics to conduct such an evaluation. We propose a new video-based benchmark for conditional single image evaluation over datasets spanning different scenes and primitives. A single frame from each video is designated for training, where the network is trained to map the primitive image to the designated training frame. The trained network is then used to map from primitive to image for all the other frames in the video and compute the prediction error using LPIPS (Zhang et al., 2018) and fidelity using SIFID (Shaham et al., 2019). We use all 16 video segments from the Cityscapes dataset (Cordts et al., 2016) provided by the code in vid2vid (Wang

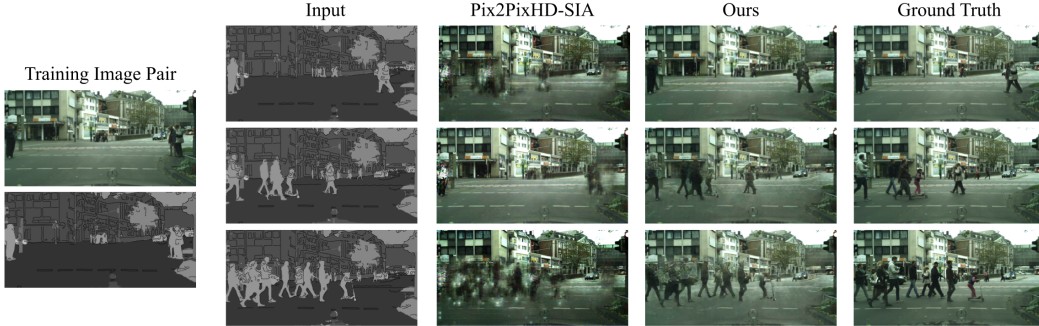

Figure 6: Several sample results from the Cityscapes dataset. We train each model on the segmentation-image pair on the left. We then use the models to predict the image, given the segmentation maps (second column from left). Our method is shown to perform very well on this task, generating novel configurations of people not seen in the training image.

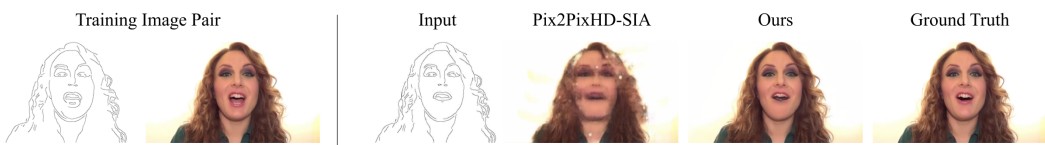

Figure 7: Comparing Pix2PixHD trained on a single image pair with the "flip-and-crop" warp vs. DeepSIM. Our method outputs an image much more similar to the ground truth. Also see App. D

et al., 2018) and Few-shot-vid2vid (Wang et al., 2019). These sequences are labelled aachen-000000 to aachen-000015 leftImg8bit. For each sequence, we train on frame 000000 and test using frames 000001 to 000029. We use the segmentation maps provided as image primitives. We also use the first 5 videos in the public release of the *Oxford-BBC Lip Reading Sentences 2* (LRS2) dataset containing videos of different speakers. We extract their edges using a Canny edge detectorCanny (1986). In total, our evaluation set contains 464 Cityscapes frames and 239 LRS2 frames.

A visual evaluation on a few frames from the Cityscapes dataset can be seen in Fig. 6. We compare our method to the results of Pix2PixHD-SIA, where "SIA" stands for "Single Image Augmented" e.g. a Pix2PixHD model that was trained on a single image using random crop-and-flip warps but not TPS. We can observe that our method is able to synthesize very different scene setups from those seen in training, including different numbers and positions of people. Our method outperforms significantly in terms of fidelity and quality than Pix2PixHD-SIA indicating that our proposed TPS augmentation is critical for single image conditional generation.

Quantitative evaluations on Cityscapes and LRS2 are provided in Tab. 2 and Tab. 3. We report LPIPS and SIFID for each of the 5 LRS2 sequences and also for the average of 16 Cityscapes videos. We can observe that our method significantly outperforms Pix2PixHD-SIA in all comparisons.

## 5 ANALYSIS

*Why use super-primitive(SP)?* The choice of different input primitives conveys different information. Segmentations capture high-level aspects of the image while edge maps capture the low-level of the image better. Pix2PixHD proposed combining instance and semantic segmentation maps, however, this does not provide low-level details. We propose a new primitive, "super primitive" (SP), that combines their strengths. Fig. 2 compares the three primitives. The edge representation is unable to capture the eye, presumably as it cannot capture its semantic meaning. The segmentation is unable to capture the details in the new background regions creating a smearing effect. SP is able to capture the eye as well as the low-level textures of the background region, showing its strong representational power. In Fig. 3 we present more manipulation results using SP. In the center column, we present

Table 2: Results for the Cityscapes dataset - we report the average over the 16 videos.

| Metric | Baseline: Seg, Crop+Flip | Ours: Seg, TPS | Ours: SP, TPS |
|---|---|---|---|
| LPIPS | 0.342 | 0.216 | **0.134** |
| SIFID | 0.292 | 0.127 | **0.104** |

Table 3: Results of Pix2PixHD-SIA (crop-and-flip) and our method (TPS) on 5 LRS2 videos (both trained on a single pair). For each sequence left column: LPIPS, right column: SIFID.

| Method | S1(L/S) | | S2(L/S) | | S3(L/S) | | S4(L/S) | | S5(L/S) | |
|---|---|---|---|---|---|---|---|---|---|---|
| Pix2PixHD | 0.44 | 0.51 | 0.47 | 0.49 | 0.41 | 0.5 | 0.53 | 0.26 | 0.46 | 0.44 |
| Ours - no VGG | 0.14 | **0.05** | 0.26 | **0.11** | 0.11 | 0.07 | 0.28 | 0.14 | 0.19 | 0.08 |
| Ours | **0.12** | 0.07 | **0.21** | 0.12 | **0.1** | **0.04** | **0.22** | **0.12** | **0.14** | **0.06** |

image reorganization results, where the positions of the rightmost cars were switched. As the objects were not of the same size, some empty image regions were filled using small changes to the edges. A more extreme result can be seen in the rightmost column, the car on the left was removed. This created a large empty image region. By filling in the missing details using edges, our method was able to successfully complete the background. An ablation is presented in App. E, showing the superiority of the SP representation. More comparisons are presented in App. D.

*Is the cGAN loss necessary?* We evaluated removing the cGAN loss used by our method (and Pix2PixHD), and using a VGG perceptual loss (the difference between the activations of a VGG network of output and groundtruth). The results on Cars are presented in App. F, we conclude that for such high-res images the cGAN is a better perceptual loss. We experimented with the VGG loss at lower resolutions, the results are reasonable, however the images are not as sharp as the cGAN loss.

*Can methods trained on large-scale datasets generalize to infrequently occurring images?* We present examples where this is not the case. Fig. 4 showed that BicycleGAN did not generalize as well as Pix2PixHD-MI for new (in-distribution) shoes. We show that in the more extreme case, where the image lies further from the source distribution used for training, current methods fails completely. We manually labelled the semantic and instance segmentation maps of the Cars image, and input it into a Pix2PixHD pre-trained by the authors on the Cityscapes dataset which contains street scenes of cars, roads and building. The results are presented in Tab. 4. We can see that a state-of-the-art method pre-trained on a large dataset does not generalize well to out-of-distribution inputs whereas our method, which is trained on a single image can perform well in such cases.

*Augmentation for single image manipulation:* Although we are the first to propose single-image training for manipulation using extensive non-linear augmentations, SinGAN can be seen as implicitly being an augmentation-based approach (for unconditional generation). In the first pyramid level it creates a low-resolution high-level random image generator, by mapping noise to low-res images. The later stages can be seen as a conditional generator which learns to map low-to high resolution images. Critically, it relies on a set of "augmented" input low-res images generated by the first stage GAN. So although this is not explicit in SinGAN, we think it also uses augmentations. On the other hand, Deep Image Prior indeed does not use any form of augmentation.

| Input Semantic Seg. | Input Instance Seg. | Output | Ground Truth |
|---|---|---|---|

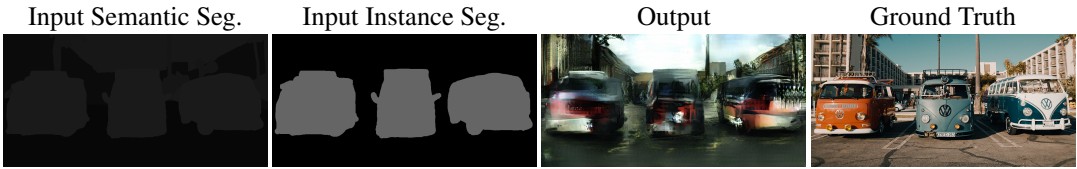

Table 4: Running a Cityscapes-trained Pix2PixHD on a primitive representaation of an out-of-distribution image. The network was not able to generalize well and generated unsatisfactory results.

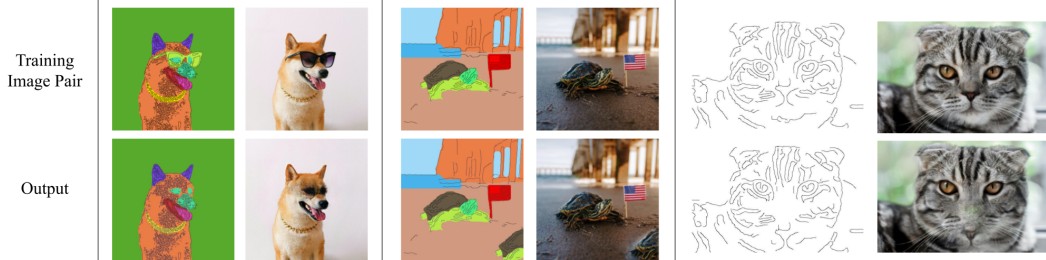

Figure 8: Failure modes: (left) generating unseen objects - eyes of the dog (center) background duplication - sea behind the turtle (right) empty space interpolation - nose of the cat

*Runtime*: As with other methods that train deep neural networks based on a single image, per-image training times are higher than when trained on large datasets. Our runtime is a function of the neural architecture we use and the number of iterations. Here specifically, we use the Pix2Pix-HD framework which has roughly comparable runtime to SinGAN for the same image size. The precise runtime depends on image size and the number of iterations used. When running all experiments on the same hardware (NVIDIA RTX-2080 Ti), smaller images e.g. the "Balloons" image showcased by SinGAN (size: 248X186) take SinGAN 50 minutes while DeepSIM (ours) takes 63 minutes. Runtime scales with the size of the image, so that large images such as the Cars image (640X320) take SinGAN 195 minutes while ours takes 185 minutes. Although this is not particularly fast, it is a general characteristic of many deep single image manipulation methods and not a particular issue of DeepSIM. We are optimistic that runtime optimizations in future work can significantly cut the runtime in all such methods, but this is not our focus.

*Failure modes:* Learning from a single image is very challenging and although our method makes encouraging progress, it has failure cases. We highlight three main failure modes (Fig. 8): i) generating unseen objects - the network is trained on just a single pair and is not aware of any other objects. When the manipulation requires generation of unseen objects, the network can do so incorrectly. ii) background duplication - when adding an object onto new background regions, the network can erroneously copy some background regions that originally surrounded the object, which may appear out-of-context. iii) interpolation in empty regions - as no guidance is given in empty image regions, the network hallucinates details, sometimes incorrectly. See App. C for further analysis.

## 6 CONCLUSION

We proposed a novel method for training conditional generators from a single training image based on thin-plate-spline augmentations. We demonstrated that our method is able to perform complex image manipulation at high-resolution. Single image methods have significant potential, they preserve image fine-details to a level not typically achieved by previous methods trained on large datasets. One limitation of single-image methods (including ours) is the requirement for training a separate network for every image, which can be expensive over a large dataset. Speeding up training of single-image generators is an important and promising direction for future work.

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

## A    ADDITIONAL RESULTS

We present additional results of our method, DeepSIM, using the SP on a range of manipulations on different images.

| Training SP | Training Image | Input SP | Output Image |
|---|---|---|---|
| | Changing the Posture of the Squirrel | | |

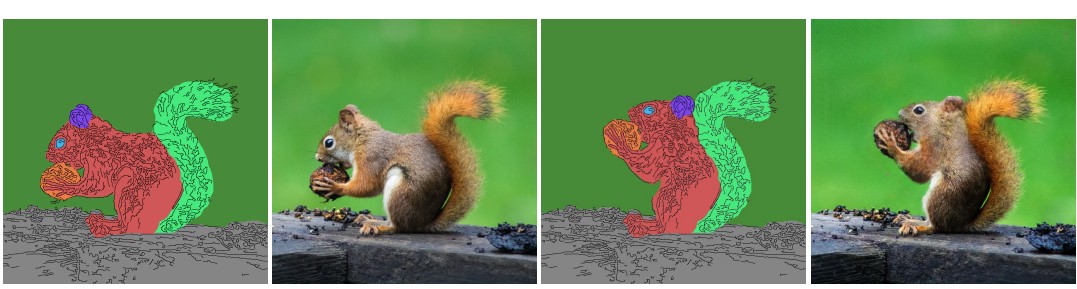

| Training SP | Training Image | Input SP | Output Image |
|---|---|---|---|
| | Changing the Shape of the Tail | | |

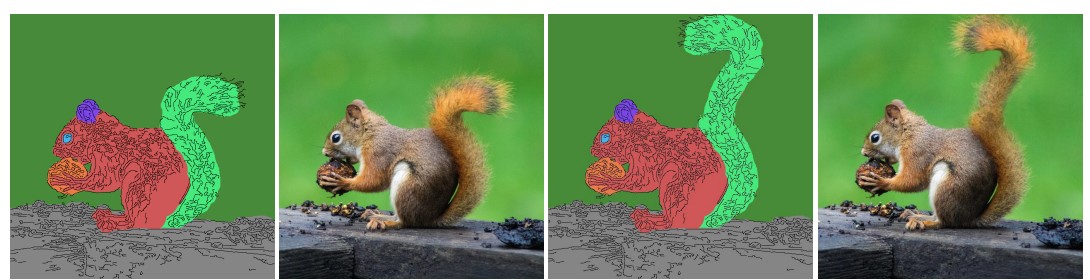

Removing the Fork

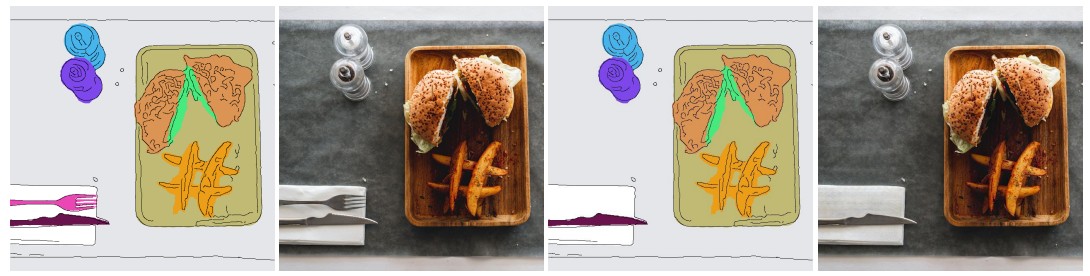

Joining the Hamburger Halves

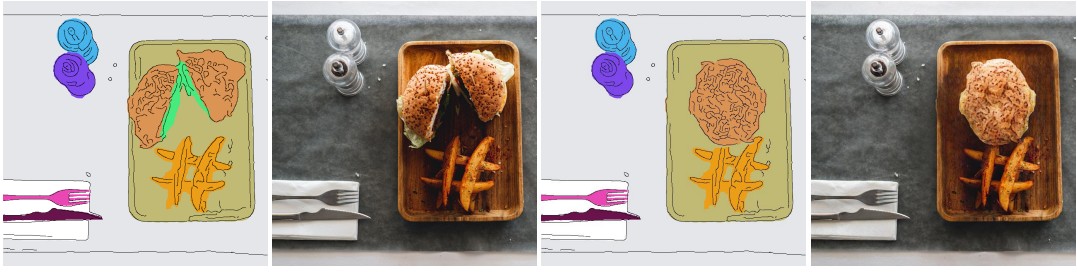

| Training SP | Training Image | Input SP | Output Image |
|---|---|---|---|

Making the Beak Longer

Changing the Position of the Wings

Removing the Handcuffs

Removing the Right Hand

| Training SP | Training Image | Input SP | Output Image |
|---|---|---|---|

Changing the Hat and Making the Face Longer

Making the Body Wider

Moving the Tree

Changing the Shape of the Tree

| Training SP | Training Image | Input SP | Output Image |
| --- | --- | --- | --- |

Changing the Shape of the Lake

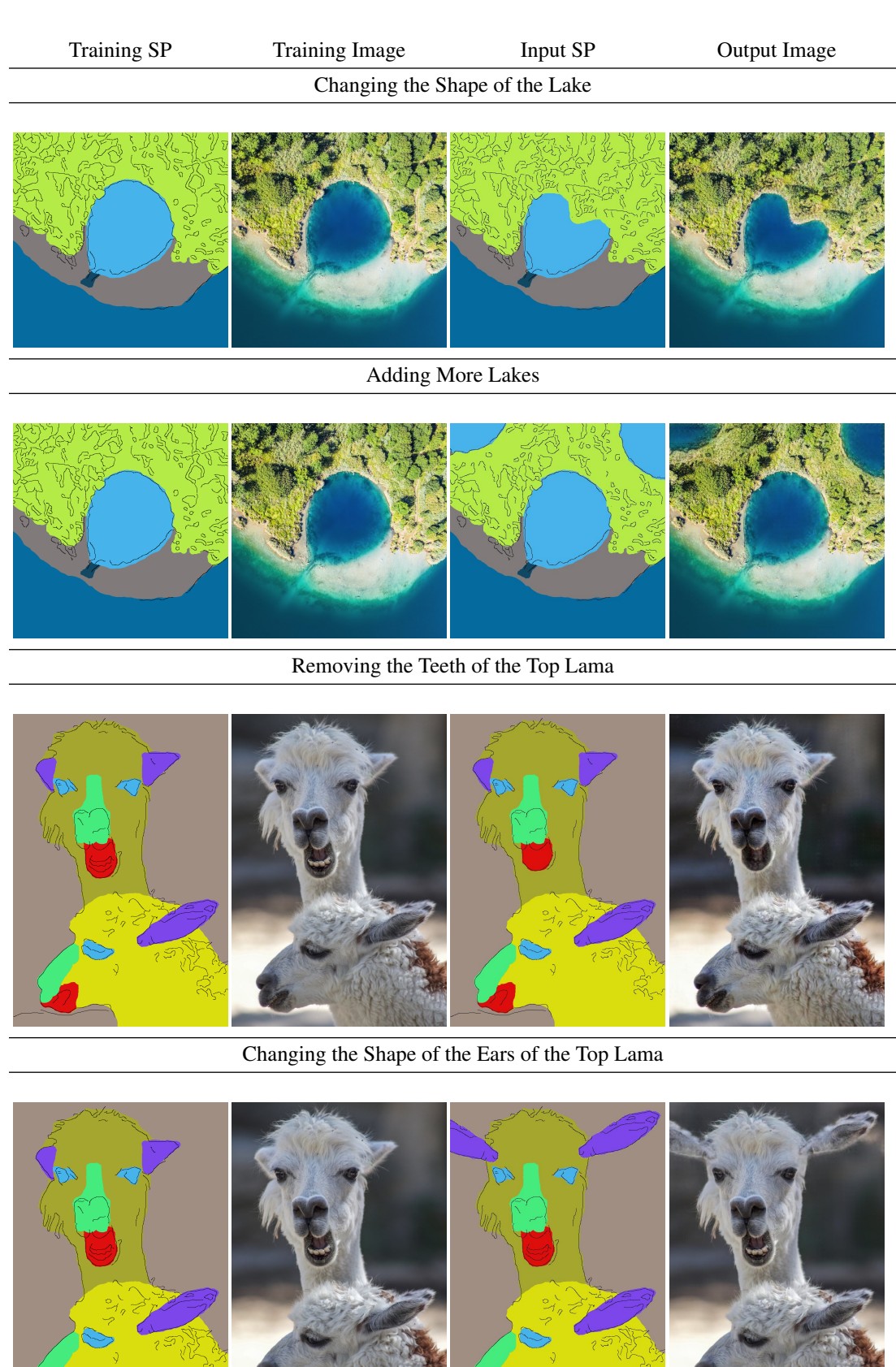

Adding More Lakes

Removing the Teeth of the Top Lama

Changing the Shape of the Ears of the Top Lama

| Training SP | Training Image | Input SP | Output Image |
| --- | --- | --- | --- |

Adding Stems

Changing the Shape of the Horns

Adding the Left Paw

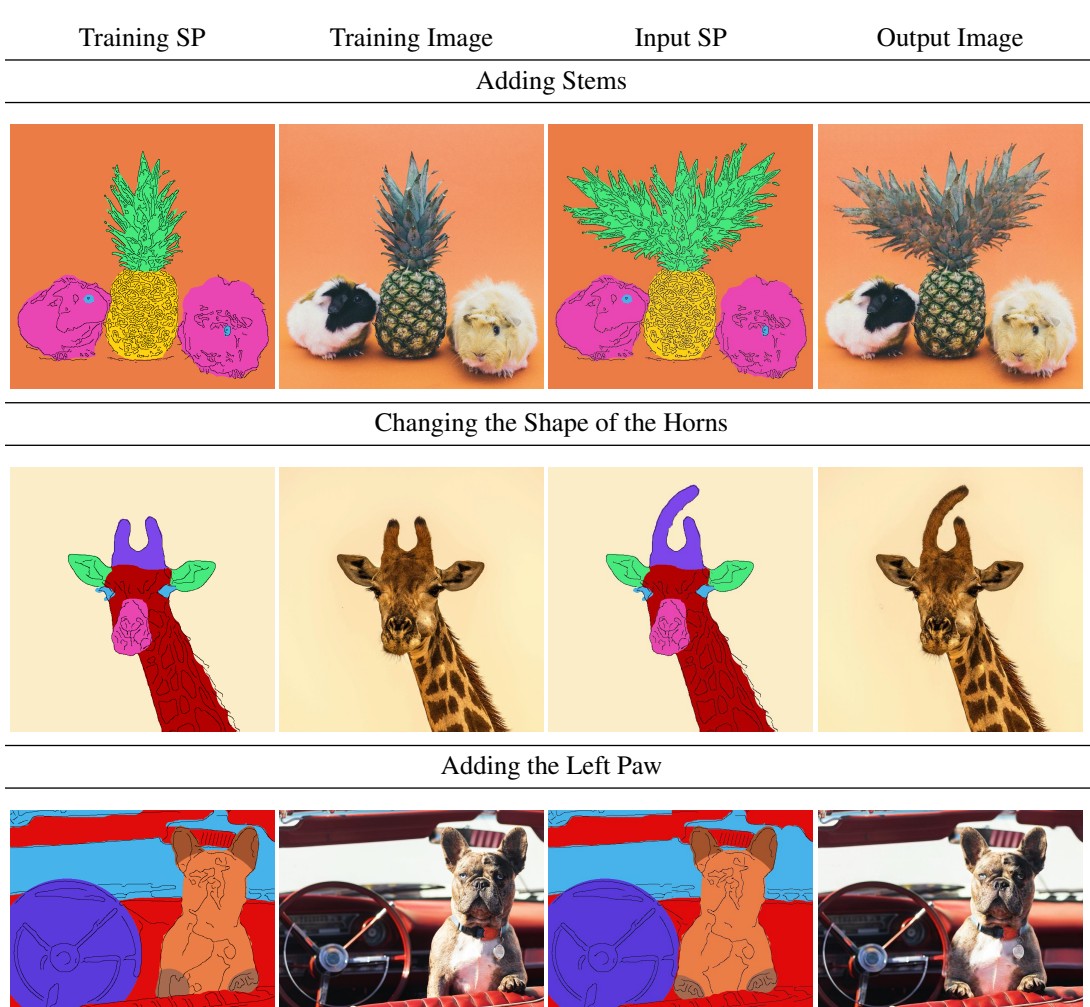

# B  A STEP-BY-STEP DEMONSTRATION OF EDITING THE PRIMITIVE

Performing complex manipulations by our method is quite easy. In this figure we present a step-by-step example of editing a primitive representation using "Paint". It simply requires sampling the required color and painting over the primitive image. One may also "borrow" edges from other areas of the image to fill in empty spaces.

| Step 1: Original Image | Step 2: Paint Heart |
|---|---|

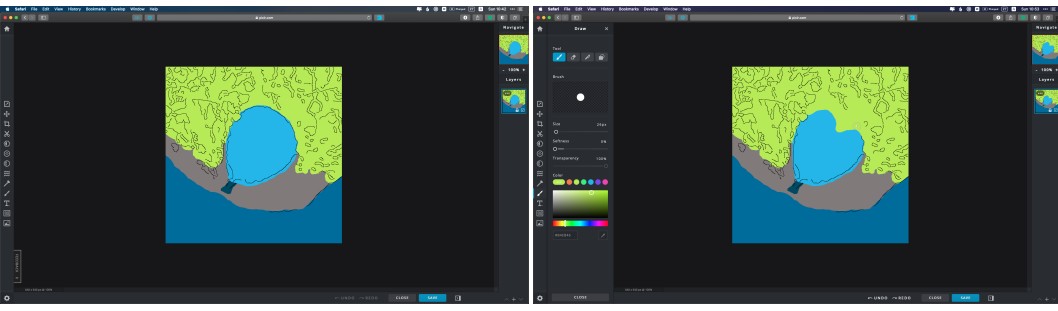

| Step 3: Copy Edges of Trees | Step 4: Rearrange Edges of Trees |
|---|---|

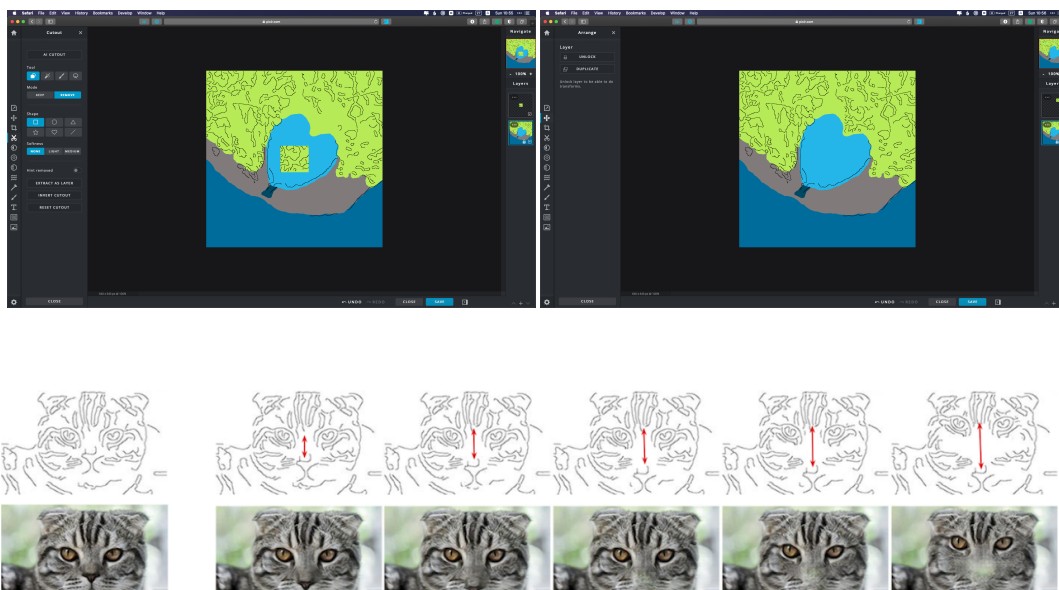

Figure 9: Evaluation of the ability of our network to interpolate across empty space regions. The two leftmost columns show the training image pair, we gradually increase the distance between the eyes and nose of the cat, and feed the test images to the network, the corresponding output of each test image is shown in the second row. Our method generates attractive interpolations for moderate changes, the performance deteriorates for larger interpolations.

## C  EMPTY SPACE INTERPOLATION

In this section we stress test our method's ability to handle regions with little guidance. In this example, the nose of the cat was shifted progressively downwards, forcing the network to interpolate the missing space. We observe the network synthesizes attractive images for moderate empty regions, however, as the empty region gets larger, the network looks for similar regions to fill the newly created void. These regions will often be areas which exhibit low amounts of detail in the primitive representation. In our case we can notice that for larger shifts, the empty space becomes greener until eventually it inpaints a background patch. We conclude that at a certain point, the network fails to learn the spatial relationship among objects in the image (i.e. that the background can not be placed on the cat's face) and satisfies the given constraint using neighboring information (as was analysed above).

# D  VIDEO FRAMES

We present a qualitative comparison between different primitives on two frames from the LRS2 datasets. Although all primitives generate surprisingly good results, given the training on just a single image, super-primitive generates cleaner outputs with fewer artifacts.

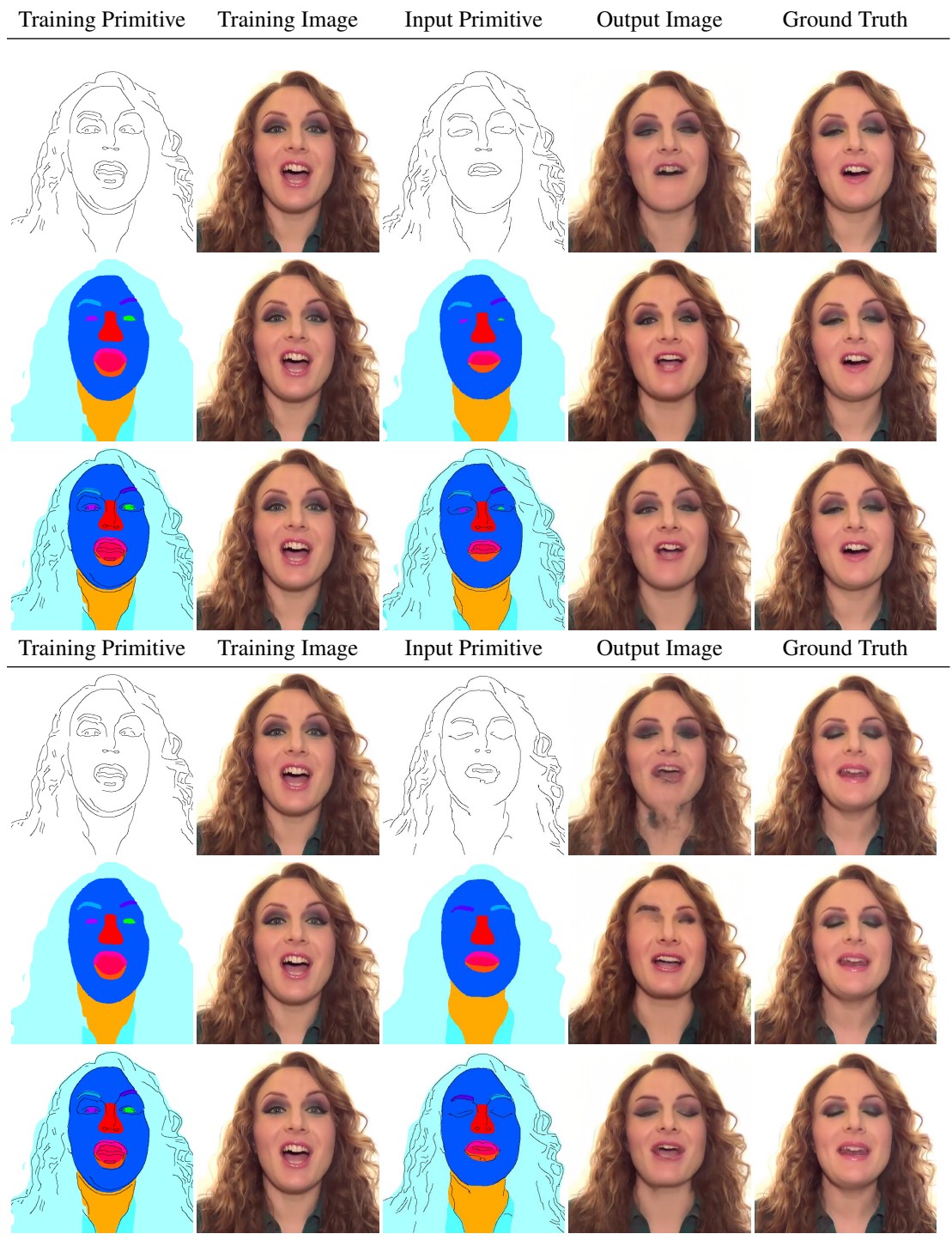

| Training Primitive | Training Image | Input Primitive | Output Image |
|---|---|---|---|

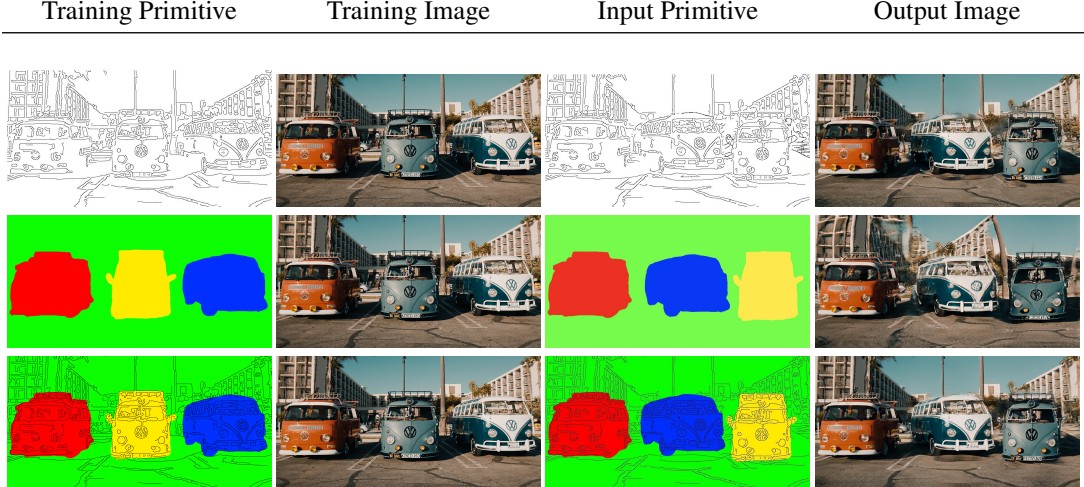

Figure 10: Ablation of SP on the cats dataset: (top) edges-only (center) segmentation-only (bottom) SP. We can see that edges-only creates wrong associations between objects, segmentation-only fails to generate the fine details correctly (e.g. building), whereas SP achieves strong results.

## E  ABLATION OF SP FOR THE CARS IMAGE

We present an ablation of the superprimitive (SP) representation for the Cars image. In Fig. 10, we present results for a manipulation on the Cars image using edge-only, segmentation-only and SP. We can see that SP generates attractive artifact free results.

## F  AN ABLATION OF THE LOSS OBJECTIVE

We compare the results of our method, DeepSIM, using the original cGAN loss as in the base Pix2PixHD architecture vs. non-adversarial losses - the simple $L_1$ loss and the perceptual loss based on the difference of VGG activations. We can see that on this image both non-adversarial losses fail. Note that at lower resolutions non-adversarial losses do indeed succeed but do not generate results of comparable sharpness of the cGAN loss. Additionally, we performed the experiment with the cGAN but without the VGG perceptual loss, the results are presented below. It can be seen that without the VGG loss, the results are reduced in quality and contain grainy artifacts.

| Training | Perceptual Loss | L1 Loss | Ours w/o VGG | Ours w/ VGG |
|---|---|---|---|---|

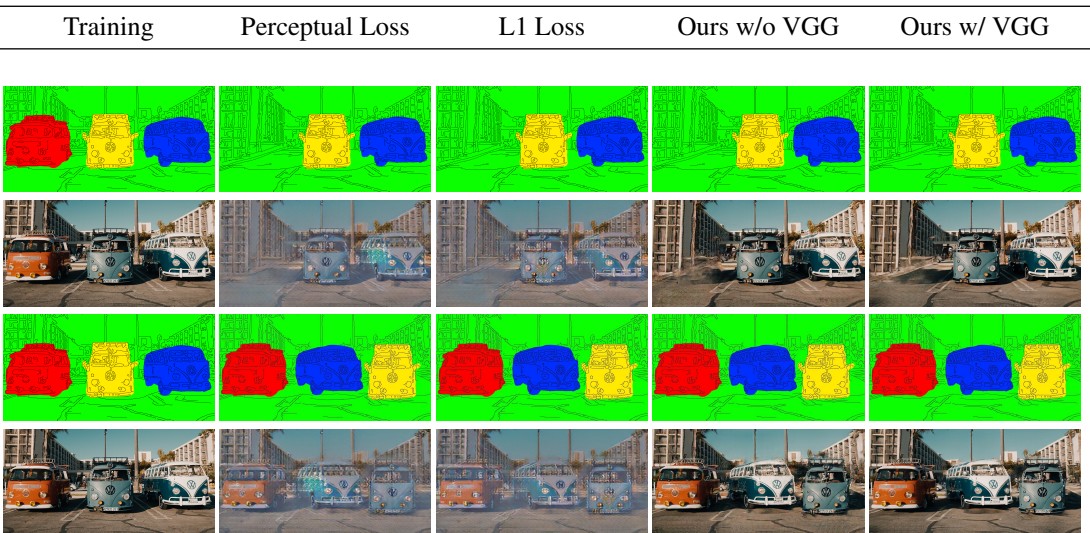

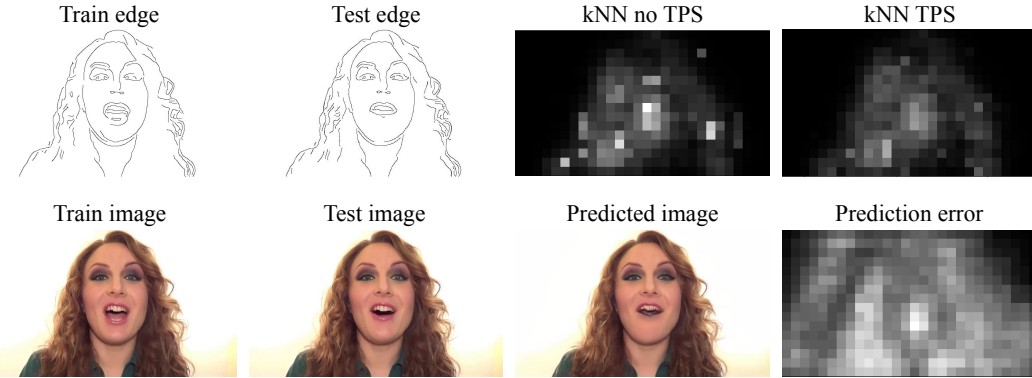

Figure 11: An analysis of the benefits of TPS. We show the kNN distance between patches in the test and train frames with and without TPS augmentations (top-right). We can see that TPS augmentation decreases the kNN distance, in some image regions the decrease is drastic suggesting the patches there can be obtained by deformations of training patches. The kNN-TPS distance appears to be correlated with the regions where the prediction error of our method is large. This analysis suggests that by artificially increasing the diversity of patches, single-image methods can generalize better to novel images.

## G TPS GENERALIZATION IMPROVEMENT

Let us consider the train and test edge-image pairs presented in Fig. 11. We input each edge map through an ImageNet-trained ResNet50 network and computed the activations at the end of the third residual block. For each pixel in the activation grid of the test image, we computed the nearest-neighbor (1NN) distance to the most similar activation of the train image. We then performed 50 TPS augmentations to the training image, and repeated the 1NN computation with the training set now containing the activations of the original training image and its 50 augmentation. Let us compare the 1NN distances presented in Fig. 11 with and without TPS augmentations. Naturally, the 1NN distance decreased for the TPS-augmented training set due to its larger size. More interestingly, we can see that several face regions which prior to the augmentations did not have similar patches in the input, now have much lower distance (while more significant changes might not be possible to describe by TPS). In Fig. 11, we present the results of our method when trained on the training edge-image pair (shown in the leftmost column) and evaluated on the test edge. We can see that the prediction error ($L_1$ difference between ResNet50 activations of the predicted and the true test image) appears to be strongly related to the 1NN distance with TPS-augmentations. This gives some evidence to the hypothesis that the network recalls input-output pairs seen in training. It also gives an explanation for the effectiveness of TPS training, namely increasing the range of input-output pairs thus generalizing to novel images.

## H TPS EXAMPLES

We present several examples of original and TPS augmented images and primitives. We can see that TPS introduces complex deformations to the samples, allowing much more expressive edits than when using simple "flip and crop" augmentations.

| Original | TPS 1 | TPS 2 | TPS 3 |
| --- | --- | --- | --- |

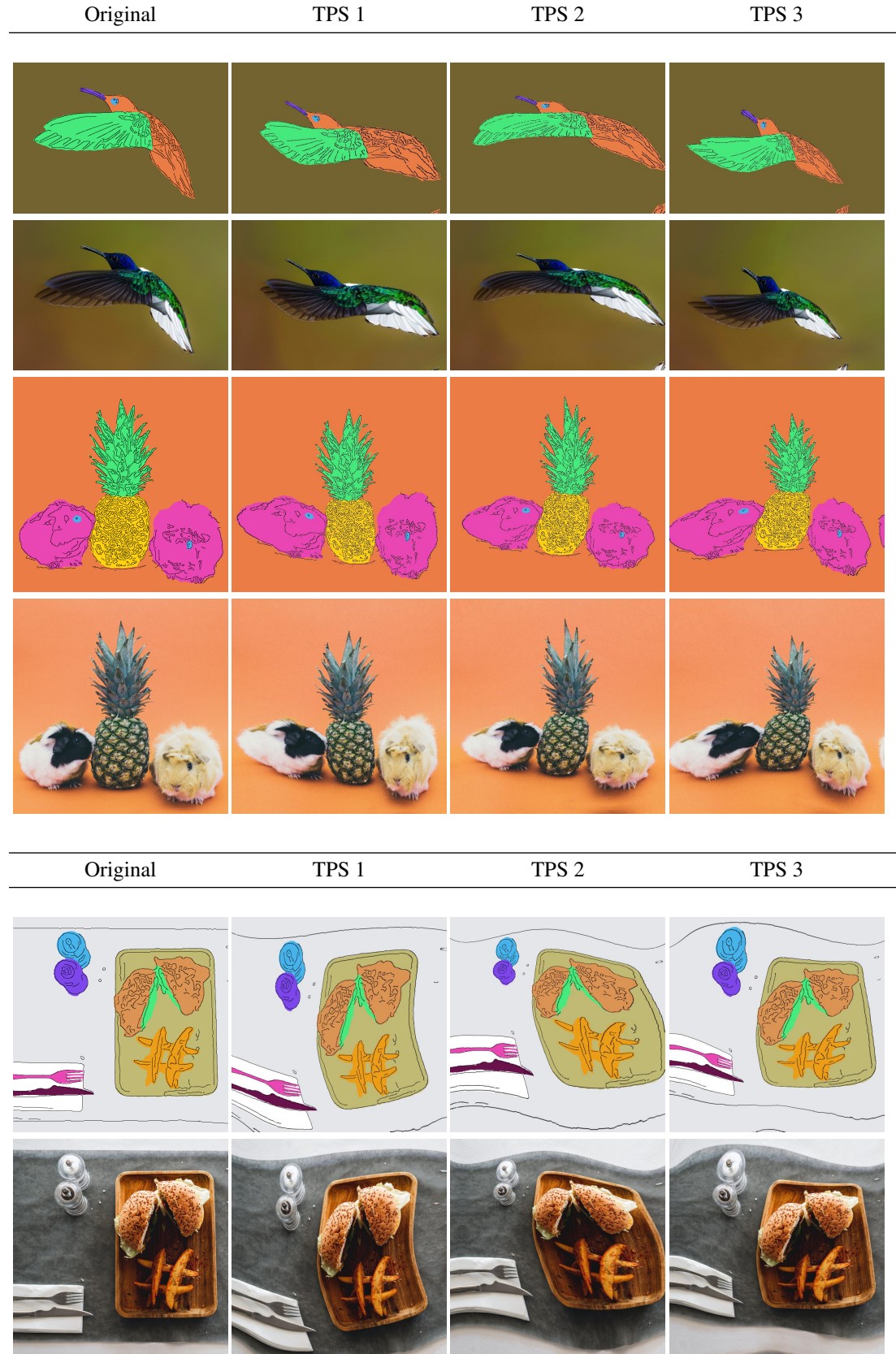

| Original | TPS 1 | TPS 2 | TPS 3 |
| --- | --- | --- | --- |

