# OpenReview forum: "Deep Single Image Manipulation"
_ICLR.cc/2021/Conference — Reject_

### Official Review · AnonReviewer4 · 2020-10-27
**Interesting take on single-image generative models**

**Rating:** 7
**Confidence:** 4

**Review:**

<Paper Summary>
This work proposes a method to design conditional generative models based on a single image. In particular, while some recent models have enabled one to sample (unconditionally) images from a generative model learned from a single image (like SinGAN), this work explores a way of conditioning the generation on a primitive, which can be user-specified. As a result, one can produce realistic modifications to a given image by modifying - or sketching - some primitive.

<Review Summary>
I like the simplicity and the ingenuity of the approach. This reviewer is not aware of any method that can produce similar results, and as such it represents the state of the art for deep-learning based single-image manipulation. At the same time, clarity (of both writing and technical aspects) could be significantly improved.

<Details>
Strengths:
* Novel formulation to train 'single' image generative models
* Flexible framework
* Compelling experimental results

Weaknesses:
* While it is true that this model trains a generative model from a single image, it does so by deploying traditional large-scale learning on many modified versions of a single input image. This is in contrast to other methods (DIP, SinGAN) that train completely on *a single image*. This distinction could be made clearer in the text.
* Many comments and expressions are too vague, making it difficult to fully understand the approach: In GAN models, the discriminator, $D$, is typically a deep-network model parametrizing a function from the space of images to the reals, say $R^n \rightarrow [0,1]$, representing the probability that the input comes from the distribution of real images as opposed to from a synthetic generator, $G$. In Eq. (2), however, the discriminator model seems to receive two outputs, $(x,G(x))$, which makes little sense to me. Did the authors perhaps meant to write simply $D(y)$ and $D(G(x))$ in Eq. 2? More broadly, this confusion comes from a lack of a clear definition of the employed functions.
* Also in Eq. (2), GANs typically have a large collection of training samples and so writing the expectation over the distribution of images $p_{data}$, makes sense. In this case, however, one has only 1 sample. It's true that the authors are artificially creating a distribution around the given sample, so perhaps they could make this more precise and clarify what they refer to as $p_{data}$.
* It would significantly help the presentation to define the different quantities and spaces used by the authors. For example, consider defining the domain and codomain of the warp $f$, which they employ to generate the augmentation. It is also not totally clear how these transformations are sampled (the $t(i,j)$'s). Later in Sec. 3.4 the authors mention that they randomly sample a new TPS wrap, but sampled how and from what distribution?
*  There's no comment as to how computational intensive the method is: How many new samples (augmentations) are generated for every image? How long does it take to generate these, and how long does the subsequent training take?
* "The long tail of images" doesn't mean anything to this reviewer. It is clear that the authors intend to refer to images that occur very infrequently in the distribution of real images, but the authors should make this more precise and avoid comments of the sort of "primitive from the long-tail", which make no sense.

---

> ### Author Response · Authors · 2020-11-16
> **Author Response**
>
> Thank you for the dedicated review. We were glad to hear the reviewer appreciated the “ingenuity of the approach” and found the experiment compelling. The reviewer made many helpful suggestions which have been incorporated in the revision.
>
> **“While it is true that this model trains a generative model from a single image, it does so by deploying traditional large-scale learning on many modified versions of a single input image. This is in contrast to other methods (DIP, SinGAN) that train completely on a single image”**: We actually view the SinGAN method as implicitly being an augmentation-based approach. In the first pyramid level it creates a low-resolution high-level random images generator, by mapping noise to low-res images. The later stages can be seen as a conditional generator which learns to map low to high-resolution images. Critically, it relies on a set of “augmented” input low-res images generated by the first stage GAN. So although this is not explicit in SinGAN, we think it also uses augmentations. DIP indeed does not use any form of augmentation. We added this discussion to Sec.5
>
> **“Did the authors mean to write simply $D(y)$ and $D(G(x))$”**: As our discriminator is conditional it takes as input both the input image and expected or generated results, we write $D(x, y)$ and $D(x, G(x))$. We clarified the text in Sec.3.1 to reflect this.
>
> **“It is also not totally clear how these transformations are sampled”**: For each new warp, we start from an equispaced $4 \times 4$ grid, we randomly shift each point by a uniformly distributed shift (the maximal shift is $0.1$ of the image). This transformation is named $t$ s.t. $(i’, j’) = t(i, j)$. We then smooth the transformation using TPS, the resulting transformation is named $f$ and has the same domain and codomain as $t$. $f$ is now applied to warp both primitive $x$ and image $y$. We added a more detailed description to Sec.3.2
>
> **Suggestions for improving notations and terms**: Thank you for many helpful suggestions, we have added them to the revision. Please let us know if further changes are needed.
>
> **Runtime**: As mentioned in Sec.6, similarly to other methods that train deep neural networks based on a single image, per-image training times are higher than when trained on large datasets. Our runtime is a function of the neural architecture we use and the number of iterations. Here specifically, we use the Pix2Pix-HD framework which has roughly comparable runtime to SinGAN for the same image size. The precise runtime depends on image size and the number of iterations used. When running all experiments on the same hardware (NVIDIA RTX-2080 Ti), smaller images e.g. the “Balloons'' image showcased by SinGAN (size: 248X186) take SinGAN 50 minutes while DeepSIM (ours) takes 63 minutes. Runtime scales with the size of the image, so that large images such as the Cars image (640X320) take SinGAN 195 minutes while ours takes 185 minutes. Although this is not particularly fast, it is a general characteristic of many deep single image manipulation methods and not a particular issue of DeepSIM. We are optimistic that runtime optimizations in future work can significantly cut the runtime in all such methods, but this is not our focus. We added this discussion to the analysis in Sec.5.

---

> > ### Comment · AnonReviewer4 · 2020-11-24
> > **Comment on responses**
> >
> > Thank you for the careful consideration and responses to my comments, which have clarified some points. The new notation, definitions and comments are more succinct and precise. I have modified my rating accordingly.

---

### Official Review · AnonReviewer3 · 2020-10-28
**Concerns on super primitive generation, technical contribution and qualitative evaluation**

**Rating:** 5
**Confidence:** 4

**Review:**

This paper provides an augmentation method to enable single image training. The network learns to map between a primitive representation of the image (e.g. edges and segmentation) to the image itself. During manipulation, the generator allows for making general image changes by modifying the primitive input representation and mapping it through the network.
On the positive side:
- The paper proposes an interesting mechanism to train conditional generators from a single image.
- The proposed super primitive works well for single image manipulation tasks.
- Some good image editing results are shown in the experiments.
On the negative side:
- The method requires a professional editing ability for editing edges of a super primitive. The generation of primitives also highly depends on the accuracy of semantic segmentation. If the segmentation is done manually, the editing process maybe time-consuming.
- The technical contribution is limited. It seems that the kernel of the framework is a direct use of cGAN without introducing many new ideas.
- The training speed is not reported. The reviewer thinks that a fast training process is important for single image manipulation.
- The qualitative evaluation is not very convincing. It’s better to conduct a user study.

---

> ### Author Response · Authors · 2020-11-16
> **Author Response**
>
>
> We thank the reviewer for the review and for recognizing the interesting mechanism and strong results. We acknowledge the concerns of the reviewer but believe they can be easily addressed:
>
> **Method requires a professional editing ability**: The edges can be simply manipulated by copy/paste and simple geometric operations such as rotate or shear, a step-by-step process is shown in App.B. This does not require a professional, and if our method becomes a product, GUI tools for making this task foolproof can be easily developed.
>
> **"If the segmentation is done manually, the editing process may be time-consuming”**: The segmentation is not particularly time-consuming, we included a new video showing it done in about a minute - which we think is quite reasonable. Again, if this becomes a product, tools for speeding this up can be easily developed.
>
> **Technical contribution**: While we recognize that evaluating how interesting a technical contribution is, is very subjective, we believe we present two interesting technical contributions i) providing an effective way of turning a single image into an entire dataset, allowing for standard and very effective kernels such as Pix2Pix-HD to operate on them successfully as if a large dataset were available ii) the super-primitive is new and the reviewer indicated that it works well.
>
> **Runtime**: As mentioned in Sec.6, similarly to other methods that train deep neural networks based on a single image, per-image training times are higher than when trained on large datasets. Our runtime is a function of the neural architecture we use and the number of iterations. Here specifically, we use the Pix2Pix-HD framework which has roughly comparable runtime to SinGAN for the same image size. The precise runtime depends on image size and the number of iterations used. When running all experiments on the same hardware (NVIDIA RTX-2080 Ti), smaller images e.g. the “Balloons'' image showcased by SinGAN (size: 248X186) take SinGAN 50 minutes while DeepSIM (ours) takes 63 minutes. Runtime scales with the size of the image, so that large images such as the Cars image (640X320) take SinGAN 195 minutes while ours takes 185 minutes. Although this is not particularly fast, it is a general characteristic of many deep single image manipulation methods and not a particular issue of DeepSIM. We are optimistic that runtime optimizations in future work can significantly cut the runtime in all such methods, but this is not our focus. We added this discussion to the analysis in Sec.5.
>
> **Qualitative evaluation**: As our setting is novel, we incorporated creative ways of evaluating it against other methods that were relevant in some settings and did not just rely on qualitative presentation of results e.g. pre-trained edges2shoes methods in Fig.4, vs. SinGAN for the tree in Fig.5, pre-trained cityscapes in Tab.4 and two quantitative evaluations in Tab.2-3. We have also included a very large number of images in the paper and appendix, which all the reviews found compelling and realistic. As the reviewer requests it, we can include in the final version a user study measuring how realistic users view the manipulation vs. the original images, and how reflective the manipulated images are of the primitive image.

---

> > ### Author Response · Authors · 2020-11-24
> > **User Study**
> >
> > We made an extra effort to address the reviewer’s concern by conducting a user study. Our protocol is similar to that of Pix2Pix and SinGAN - we sequentially presented 30 images: 10 real, 10 manipulated images, and 10 of side-by-side pairs of real and manipulated images. The participants were asked to classify each as “Real” or “Generated by AI”. In the case of pairs, we asked participants to determine if the ‘left’ or ‘right’ image was real. Each image was presented for 1 second, in accordance to previous protocols. The study consisted of 140 participants. (104 males, 36 females). We used Google Forms for this study, as this is faster to set up. The confusion rate on the unpaired images was 42.6%, while on the paired images it was 32.6%. This shows that our results are very realistic and the manipulated images are hard to tell apart from real ones. We have preliminary results for the case, where presentation of images is not time-limited, the confusion rates are a little lower (about 4%), we will further investigate this setting in future work.

---

> > > ### Comment · AnonReviewer3 · 2020-11-25
> > > **Not convinced to accept the paper**
> > >
> > > Thanks for the response and the user study. Unfortunatelly, I am still not convinced to accept this paper due to the limited technical contribution. I agree that the super-primitive proposed in the paper is interesting, however, the network and the loss function are both somethat direct use of existing works. I don't think the current technical contributions are important enough to be accepted by ICLR. Thus, I keep my previous rating.

---

### Official Review · AnonReviewer1 · 2020-10-29
**The proposed method in this paper demonstrates its effectiveness in single image manipulation task through various results, including both quantitative and qualitative experiments. However, additional descriptions on the effectiveness of the objective function and a primitive representation are required to make this paper convincing. Therefore, I vote for ‘Marginally above acceptance threshold” for this submission, but I may reconsider my assessment if all the concerns are resolved.**

**Rating:** 6
**Confidence:** 4

**Review:**

This paper proposed a single image-based manipulation method (DeepSIM) using conditional a generative model. The authors addressed this problem by proposing to learn the mapping between a set of primitive representation, which consists of edges and segmentation masks, and an image. They also adopted a thin-plate-splines (TPS) transformation as augmentation which enables the model to robustly manipulate an image by editing primitives.

Pros
-	This paper is clearly written and easy to follow.
-	The authors proposed a novel conditional manipulation method based on a single image, which is new in this area.
-	DeepSIM is capable of generating the plausible results by manipulating its contents in both a low and a high-level manner, maintaining its realism and fidelity.

Cons
-	The authors need to clarify why the VGGNet-based perceptual loss encourages the model to maintain the fidelity. As Johnson et al. [1] argued that this loss is defined as humans’ perceptual difference between the images, it would be helpful to further explain how the model with this loss between G(x) and y better reflects the primitive representations in the generated output than the model without it. Additional explanation and experiments, such as ablation study, would make the paper convincing.
-	Compared to the segmentation, the edge map seems to less contribute to the image manipulation. Most of the results are mainly attributed to the segmentation changes, and only slight modification is caused by the change of edge primitive, as shown in Appendix. A “Removing the teeth of the top lama.” Additional qualitative results for drastic manipulation in a low-level manner caused by the edge primitive would be necessary.
-	I think the technical novelty of TPS-based augmentation in the paper is not significant in that the TPS transformation has been widely used in existing literature [2][3] for learning correspondence between two images.
-	As the authors mentioned in the conclusion, training a network for a single image manipulation would be a critical bottleneck in practical use. In this respect, the training time on a single image should be reported. Additionally, detailed descriptions how to obtain the primitives (edge and segmentation) for the input image would be required.

[1] Johnson et al., “Perceptual Losses for Real-Time Style Transfer and Super-Resolution.”, ECCV’16

[2] Han et al., “VITON: An Image-based Virtual Try-on Network.”, CVPR’18

[3] Lee et al., “Reference-Based Sketch Image Colorization using Augmented-Self Reference and Dense Semantic Correspondence.”, CVPR’20

-------------------------------------
After rebuttal:

Thank you for the dedicated consideration of my comments, but there are a few remaining concerns that are not clear.

1. The editing effects of edge maps are not distinct from those of segmentation maps. More specifically, except for the background in Fig.3 and the second face in App.D, most of the examples demonstrate the same kinds of manipulation as shown in the samples of the segmentation map manipulations. This includes moving, stretching, and erasing the objects. I think that the qualitative results of edge modification are not sufficient to prove its effectiveness, compared to those of segmentation maps.

2. As shown in the image segmentation video the authors provide, a user needs to segment every single object which are selected for the manipulation. Moreover, segmenting small and fine objects requires further elaborate and laborious annotations from the user, resulting in a critical bottleneck for practical use.

Due to these concerns, I would keep my previous rating of “6. Marginally above acceptance threshold.”

---

> ### Author Response · Authors · 2020-11-16
> **Author Response**
>
> Thank you for the positive review and for recognizing the method’s novelty. We believe we can resolve the reviewer’s remaining concerns:
>
> **Perceptual loss**: We added a new ablation in Fig.3 and App.F demonstrating the effect of the addition of the VGG perceptual loss - it can be seen from the qualitative ablation that without VGG loss the results are a bit grainy. From the quantitative ablation, we can see that it typically improves the results on both metrics. This is in line with the ablation of the original Pix2PixHD paper  The ablation for using only the VGG perceptual loss (and removing the cGAN) was already shown in App.F, the results were not competitive with the full method.
>
> **Qualitative results for drastic manipulation in a low-level manner**: We would like to turn the reviewer’s attention to the large number of such results presented in the paper including: the face in Fig.1, the beak in Fig.2, the background regions in Fig.3, the shoes in Fig. 4, the faces in Fig. 7 and the entire LRS2 evaluation in Tab. 3, the cloud in the bottom two rows of Page 14, the left paw in the last figure of page 16. In addition we provide ablation study of the effect edges have on the manipulation in App.D.
>
> **TPS-based augmentation**: We mentioned multiple previous works that used TPS for various image alignment tasks, and cited several such works in Sec.2, but we are the first to use them for training conditional manipulation tasks from a single image. We added the references to the paper.
>
> **Runtime**: As mentioned in Sec.6, similarly to other methods that train deep neural networks based on a single image, per-image training times are higher than when trained on large datasets. Our runtime is a function of the neural architecture we use and the number of iterations. Here specifically, we use the Pix2Pix-HD framework which has roughly comparable runtime to SinGAN for the same image size. The precise runtime depends on image size and the number of iterations used. When running all experiments on the same hardware (NVIDIA RTX-2080 Ti), smaller images e.g. the “Balloons'' image showcased by SinGAN (size: 248X186) take SinGAN 50 minutes while DeepSIM (ours) takes 63 minutes. Runtime scales with the size of the image, so that large images such as the Cars image (640X320) take SinGAN 195 minutes while ours takes 185 minutes. Although this is not particularly fast, it is a general characteristic of many deep single image manipulation methods and not a particular issue of DeepSIM. We are optimistic that runtime optimizations in future work can significantly cut the runtime in all such methods, but this is not our focus. We added this discussion to the analysis in Sec.5.
>
> **Description of how to obtain the primitives**: The edges are computed using Canny edge detector. For segmentations, one may use existing semantic segmentation methods if relevant. In case there are no existing methods which provide accurate enough segmentations for a particular image, one may perform manual segmentation, we included a new video showing it done in about a minute - which we think is quite reasonable. If this becomes a product, tools for speeding this up can be easily developed.
>
> We hope all the reviewer’s questions were addressed and hope the reviewer will indeed reconsider changing the score.

---

### Decision · Program_Chairs · 2021-01-07
**Final Decision**

**Decision:**

Reject

**Comment:**

The reviews are a bit mixed. While all the reviewers feel that the paper proposed an interesting mechanism to train conditional generators from a single image and demonstrated good image editing results in the experiments, there are also common concerns about the practicality of the proposed method for interactive image editing. All the reviewers asked for the computation time, and some expressed the concerns about technical contributions. While these concerns were (somewhat) addressed in the rebuttal, the AC feels that it’s a hard sell to bet on the dramatic increase of computational capacity to make the computing time from an hour to realtime. Concerns about novelty also remained. Given the drawbacks, the final decision was to not accept. However, this work is promising and can be made stronger for publication in a later venue.